# 2-D Selector Simulation Studies on Grain Selection for Single Crystal Superalloy of CM247LC

**DOI:** 10.3390/ma12233829

**Published:** 2019-11-21

**Authors:** Hang Zhang, Xintao Zhu, Fu Wang, Dexin Ma

**Affiliations:** 1State Key Laboratory for Manufacturing System Engineering, School of Mechanical Engineering, Xi’an Jiaotong University, Xi’an 710049, Shaanxi, China; zhanghangmu@mail.xjtu.edu.cn; 2Foundry Institute, RWTH Aachen University, Intzestrasse 5, 52072 Aachen, Germany; zhudb8@gmail.com (X.Z.); d.ma@gi.rwth-aachen.de (D.M.)

**Keywords:** grain selector, single crystal, directional solidification, CM247LC

## Abstract

In the present work, the single crystal superalloy CM247LC was selected as the research material. By using directional experiments and the cellular automaton finite element (CAFE) model, the process of grain texture evolution in a two-dimensional grain selector was investigated to clarify the mechanism of grain selection in the two-dimensional passage during the process of directional solidification (DS). To optimize single crystal turbine blade production processes, the effects of grain selector geometries (i.e., selector diameter and pitch length, take-off angle) on the microstructure and stray grain were simulated and discussed.

## 1. Introduction

Ni-based components are used widely in the hot ends of aero-engines and gas industrial turbines (GITs) due to their excellent performance at high temperatures [1]. SX (Single Crystal) components are conventionally produced by using the directional solidification process. In this process, the seeding and selector methods are employed. Although the seeding method can precisely control the orientation of the SC component, it can also cause the occurrence of casting defects, and the operation of this method is inconvenient in practice. Therefore, the selector method is usually used in industrial applications, but for some high-requirement SC casting, a combination of both methods is applied [2,3,4].

In current practice, a three-dimensional (3D) spiral selector, called a pig-tail, is always employed in the production of SC components [5]. The geometry of the selector determines the grain selecting efficiency [6]. One of the parameters of the selector is the selector diameter. The previous investigation indicates that the smaller wire diameter, the more efficient spiral selector becomes. However, if the selector diameter is too small, the spiral selector is not strong enough to support the SC casting. Also, the cost of the model for injecting the 3D selector is high and the operation process is complex. 

To simplify the operation process and production, two-dimensional (2D) selectors were designed in the present research. Using three-dimensional (3D) printing processes, these selectors were manufactured from different materials, and the efficiency of these 2D selectors were proved by utilizing them to cast SC components. The variation in the geometry of the different materials was studied [7,8]. Additionally, the grain selection and solidification microstructure of a single crystal Ni-based super alloy in these 2D selectors were investigated and simulated.

Previous studies have shown that the geometric parameters of a spiral grain selector (i.e., the parameters of the starter block and selection block) directly determine the final result of the selected crystal [9]. Esaka et al. believed that increasing the ratio of the height of the starter block and the diameter of the starter block (D) increases the acceptability of the single crystal. Meanwhile, Esaka et al. used a two-dimensional model to simulate the influence of the take-off angle of the crystal grain on the single crystal acceptability. The results showed that when the take-off angle of the die is about 40°, the single crystal processes the highest acceptability. In the low diameter range, the acceptability of the single crystal also increases with the increase of the diameter. The height of the grain selector and the diameter of the starter block have no significant effect on the final acceptability of the single crystal [10]. Dai et al. analyzed the grain growth and grain selection process of different crystal selectors through numerical simulation and experimental research. The results showed that the geometric characteristics of the crystal selector have a great influence on the crystal selection effect. With the increase of the take-off angle, the position of the single crystal in the spiral part increases gradually. The results also showed that with increasing selector diameter, the position of the single crystal structure in the helix decreases. In the case of determining the height of the crystal selector, Meng et al. analyzed the crystallization efficiency of different crystal selectors in the process of preparing single crystal. The results show that, with the increase of the take-off angle and the decrease of the diameter, the position of the single crystal in the helical segment increases gradually. The research of Seo et al. showed that with the decreasing pouring temperature of the alloy, the nucleation density on the surface of the water-cooled copper plate increases. Additionally, they believed that the acceptability of the single crystal will also increase with the increase of the nucleation rate at the bottom of the selection block. Therefore, the design of a spiral selector should mainly consider the influence of the geometry and parameters of the starter block and selection block on the quality of the single crystal so as to improve the acceptability of the single crystal and optimize the orientation of the single crystal.

Dong et al. proved that the geometrical blocking mechanism is the governing reason for the single crystal grain selection process in grain selectors, and demonstrated that it is reasonable to use a 2D model to study the influence of geometry parameters of the grain selector on the grain growth process. However, the axial texture was absent due to the range of axial orientation often deviating from the <001> direction by 12–20 degrees [11]. The relation between geometry parameters and the grain growth process has also been briefly reported: Reducing the pitch length increases the turns in the selector, which increases the likelihood of the geometrical blocking mechanism to dominate. Reducing the selector diameter allows fewer grains to enter the grain selector and also reduces the available space for grains to develop. Increasing the take-off angle allows grains to grow in a positive thermal gradient over a longer solidification height until they reach the spiral wall. In summary, both the geometrical blocking mechanism as well as thermal control mechanism are responsible for final selection [12].

However, due to the difficulties of research and analysis on 3D spiral grain selectors, and in order to gain optimal parameters for grain selectors, a 3D grain selector was projected on a 2D surface. Hence, we can analyze the optimal 2D parameter in order the optimize the 3D spiral grain selector, C-form grain selector with variant diameter and pitch length, and Z-form grain selector with variant diameter and take-off angle. Figure 1 shows a conventional grain selector, where the upper section is the spiral selector and the cylindrical base is the starter block. Recently, Dai investigated the role of 3D spiral parts on grain selection and concluded that the geometry of the spiral grain selector has a greater influence on the efficiency of the grain selection process. The efficiency of the spiral decreased considerably by increasing the take-off angle, and Dai concluded that the take-off angle should be preferred in the region of 25° to 30°. Esaka developed a 2D analytical model depending upon the theory of columnar dendrite growth. The grain selection process efficiency was studied with the 2D model in terms of the height of starter block, initial number of seeds, “pig-tail” width, take-off angle, and length. This research greatly influenced the present work since no experimental work has been conducted due to time and cost constraints.

The aim of this study was to conduct a systematic study on the grain selection process in 2D grain selectors to improve the understanding and, hence, the efficiency of the process. For this study, a series of casting trails and simulation analysis were performed with newly designed 2D grain selectors based on various take-off angles, selector diameter, and zigzag width to understand the single-crystal texture evolution occurring in the zigzag section. The dimensions were kept close to the parameters currently used in industry to maintain the consistency of the process.

In this study, a 2D grain selector was designed. A series of industrial simulations were conducted to exhibit the physical processes occurring in the selector parts with different selector’s diameter (d_w_) and pitch length (d_s_), whereas the height (h_s_) and take-off angle (θ) were kept constant, as shown below (Figure 1 and Figure 2).

The main steps are divided into several sub-steps.

The DS process is modeled by Procast (as shown in Figure 3). Firstly, input the CAD model and pre-adjust the withdrawal rate. Meshing and simulating the DS process by the CA-FD method. The thermodynamic databases can be calculated step by step. The key parameters, such as temperature gradient and paste width, are analyzed. In the simulation process, if stray particles appear on the casting, the withdrawal rate is adjusted to the ideal state, and the new withdrawal rate is recalculated in the simulation process. Finally, when there is no prediction defect in the calculation process, a withdrawal rate curve can be obtained.

## 2. Simulation Process

### 2.1. Pre-Simulation-Equipment

#### 2.1.1. Materials Choosing

MM247LC is used as the casting material in our simulation. Its chemical composition is given below (as shown in Table 1). 

#### 2.1.2. Equipment Design

Because of the new design concept, the equipment is different from the traditional HRS (High Rite Speed) directional solidification equipment. 

A self-made Bridgman modified double-zone graphite resistance heating directional solidification device was used in our simulation. It mainly includes the furnace body, vacuum system, pumping system and electric control system, power transmission system, circulating cooling water system, and so on. The main technical parameters include the highest heating temperature of the heat preservation district, upper area: 1700 °C, lower area: 1750 °C, the withdrawal speed is 3 mm/min, the ultimate vacuum is 6.6 × 10^−3^ Pa. Every part of the system is described in detail in Figure 4.

The furnace body is mainly composed of the induction melting part, the graphite resistance double zone hot part, and the cool part. The equipment can be used for high-speed directional solidification simulation and liquid metal cooling interchangeability simulation, and the shell can be strengthened and heated locally. The heating device is composed of two heating bodies. The main heating body makes the temperature of the shell reach above the melting point of the alloy in a predetermined area and maintains a certain degree of superheat.

Vacuum system: To extend the life of the heating body and stability, the working environment must ensure a higher vacuum due to the use of graphite resistance heating equipment.

Pumping system: The function of the pulling system is to pull the sample in the process of directional solidification. The pull-out mechanism is located below the cooling zone, and the pull-out rate can be controlled through the predetermined program of PLC (Programmable Logic Controller). The operating range is 2~2000 μm/s with a speed of 0–150 rpm^−1^. The pulling speed is mainly controlled by the precision servo motor, and the high precision ball screw and linear sliding guide rail are used to ensure the precision of pulling speed. The clutch adopts integral electromagnetic clutch, greatly improving the transmission accuracy, which helps to reach an accuracy of ±2 μm/mm.

The temperature of pouring was 1773 K (1500 °C), the temperature of cooling water was 313 K (40 °C), the drawing speed was 3 mm/min, and the shell was 7 mm in diameter. The material was a CM247LC single crystal superalloy.

#### 2.1.3. Grain Selector Design

The 2D grain selector consists of a selector and a start blocker. The starter size is 10 mm (L) × 10 mm (W) × 30 mm (H). The grain selector portion was designed with varying take off angle (θ = 15~55°) and thickness (d_w_ = 0.18~0.54 cm), as shown in Table 2, Table 3 and Table 4.

As shown in the picture 4, the cooling efficiency decreases with the distance from the cooling plate. The following figures (Figure 5 and Figure 6) shows the temperature field distribution and solid-liquid interface morphology at different withdrawal times under the withdrawal rate of 3 mm/min. It can be found that all isotherms are slightly “convex” at the starter of the selector. As the solidification process progresses, the liquid isotherm gradually flattens, and the isotherm of the selected segment presents a certain “depression”. It can also be seen from the figure that as the solidification process progresses, the spacing between isotherms gradually increases, and the width of paste zone rapidly widens with the solidification process. As can be seen from the figure, the shape of the temperature field and the isotherm under different diameters is almost the same, indicating that the diameter has little effect on the temperature field.

The temperature field varies with different pitch lengths. When the pitch length increases, the temperature gradient of the selected crystal segment increases, indicating that the pitch length affects the temperature field.

## 3. Simulation Results and Discussion

### 3.1. Effect of Selector Diameter on Grain Selection

The effect of the grain selector diameter on grain selection is exhibited in Figure 6. As follows, with the selector diameter increasing from 2.6 mm to 3 mm, a single crystal can be selected. As it is cited by a solid green circle. However, when it is larger than C_2_ (3 mm), the SX selection failed, as is marked by red circles in Figure 7.

When the diameter is small, where C_2_ (d_w_ = 3 mm, d_s_ = 8 mm) is chosen as an example, the grain structure is a single crystal. As shown below, three grains in different colors entered into the selector part of the selector with 3 mm diameters. At the beginning of the solidification, grain B grows faster than grain A and grain C because its dendrite tips require less cooling. Both the primary tips of grain B and the selector wall block the growth of grain C. However, as grain B is growing, the selector wall will block its growth, and the growth speed of the secondary dendrite is lower than that of grain A. As a result, only grain A survives and grows into the castings (as shown in Figure 8 and Figure 9).

When the diameter is large, both the optimal crystal (blue part) and normal crystal (red part) grow upwards, causing the stray structure. Under this condition, the grain B has enough space to germinate, and thus to overcome the disadvantage of the slow growth space. The single grain cannot be selected (as shown in Figure 10 and Figure 11).

Although the results show that the effect of selecting the smaller diameter selector is better, however, since when the diameter is smaller than 2.4 mm, the selector is in low strength and is easy to deform, a diameter of 3 mm is expected considering the stability. To sum up, 3 mm is the optimal diameter.

### 3.2. Effect of Pitch Length on Grain Selection

As follows (Figure 12), it can be seen that with a pitch length of 8 mm or larger, a single crystal can be selected. However, when it is less than 6 mm, the SX selection failed, as is circled by red.

As in the case of the selector diameter, at the beginning of the solidification, grain B grows faster than grain A and grain C due to the good alignment with the <001> direction and the lower required undercooling. Grain C grows lower than grain B. When the pitch length is short, both the optimal crystal grain B (blue part) and the crystal grain A (red part) grow upwards, causing the stray structure (as shown in Figure 13 and Figure 14). 

The growth of grain B are both blocked in larger pitch length, and the growth speed of secondary dendrite is lower than that of grain A. Only grain A can exist in the casting (as shown in Figure 15 and Figure 16). 

However, since when the pitch length is larger than 20 mm, it is easy to deform, a pitch length of 8 mm is expected considering the stability. Summarizing, 8 mm is the optimal pitch length.

### 3.3. Effect of Take-Off Angle on Grain Selection

As shown in Figure 17, single-crystal microstructure could be achieved when the take-off angle is under 40°, as it is circled in green., while stray grains exist in the grain selectors with a take-off angle larger than 40°, as is circled in red.

For the grain selector with a small take-off angle, as shown below, where the take-off angle is 15° (as shown in Figure 18 and Figure 19), when dendrites grow into the Z-form grain selector part, growth of grain B and grain C are limited by the wall except grain A, which is close to the grain selection side. Thus, the single grain structure could be achieved.

Figure 20 and Figure 21 show that when the take-off angle is 40°, the dendrites from the outward part as well as the core of the connection wire could survive. The outer dendrites and the inner core are more likely to survive. Generally, the inner core grain B will grow into the final main grain, while the outer grain is stray. In this case, the single-crystal structure is less easily to achieve.

It’s obvious that the height of the SX structure increases with the take-off angle through CAFE simulation. Due to the increase of the take-off angle, the growth route of the grain in Z channel will be reduced at the same height; the grain selection process must need more height to compensate for the selection path. On the other hand, the increasing take-off angle makes the channel steeper, which results in impeding the grain growth, weakening the effect of selection. So, the selection is less efficient. Considering both the efficiency and the difficulty in actual production, the take-off angle should be controlled between 30° and 40°.

With the same as that in the C-form selector, keeping the take-off angle as 40°, take eight groups of simulations with variant diameters. When the diameter is less than 3 mm, a single crystal is selected. However, when the diameter is larger than 3 mm, stray grain appears. The height where the single crystal is selected reduces gradually with increasing diameter from 2.6 mm to 3 mm, which means we can get an SX structure in a lower height and less time. Thus, we can develop the efficiency by a smaller diameter. The diameter of 3 mm is well desired, considering both efficiency and stability.

## 4. Conclusions 

An innovative 2D grain selector was studied using the CAFE method by Procast. The results clearly showed that for the selection behavior of the grain selection, a smaller take-off angle, a smaller diameter, and a larger pitch length were more efficient. Accordingly, a grain selector’s take-off angle of 40°, diameter of 3 mm, and pitch length of 8 mm, respectively, are recommended.

## Figures and Tables

**Figure 1 materials-12-03829-f001:**
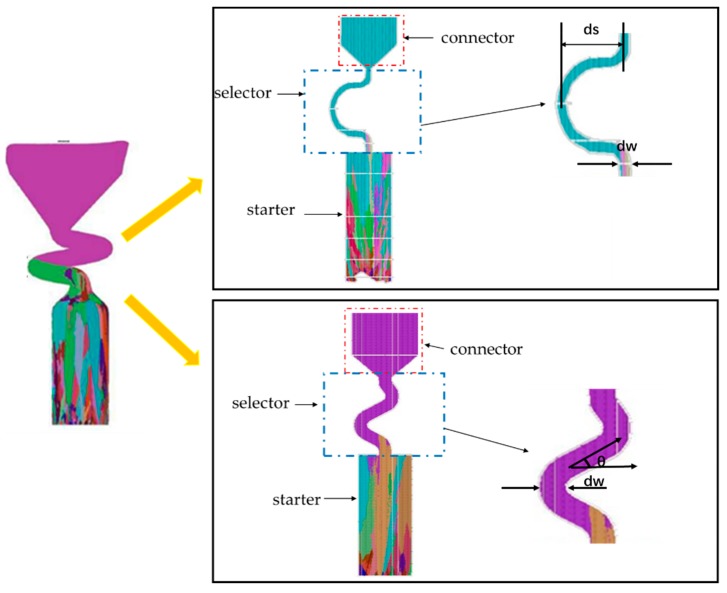
Schematic drawings of C-type and Z-type and the parameters used in the selector part.

**Figure 2 materials-12-03829-f002:**
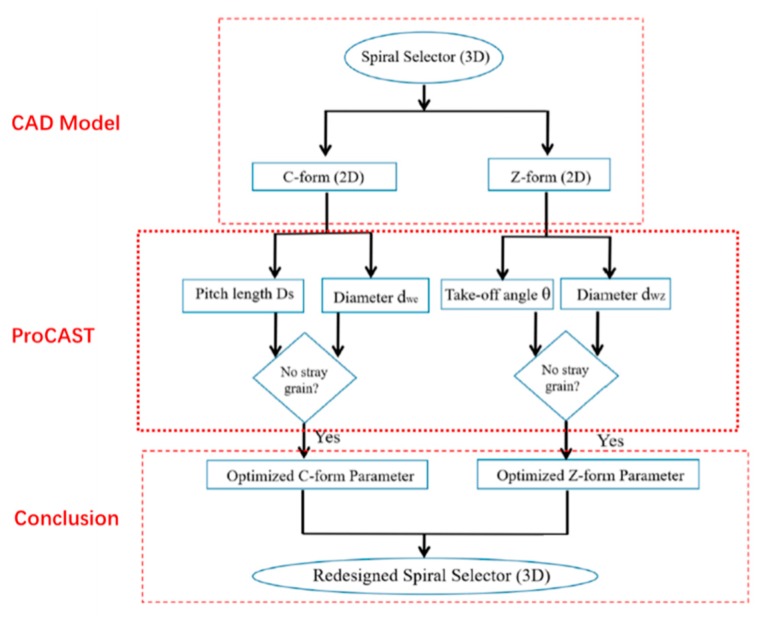
The flow chart of find a high-efficient grain selector using the reduction dimensional method.

**Figure 3 materials-12-03829-f003:**
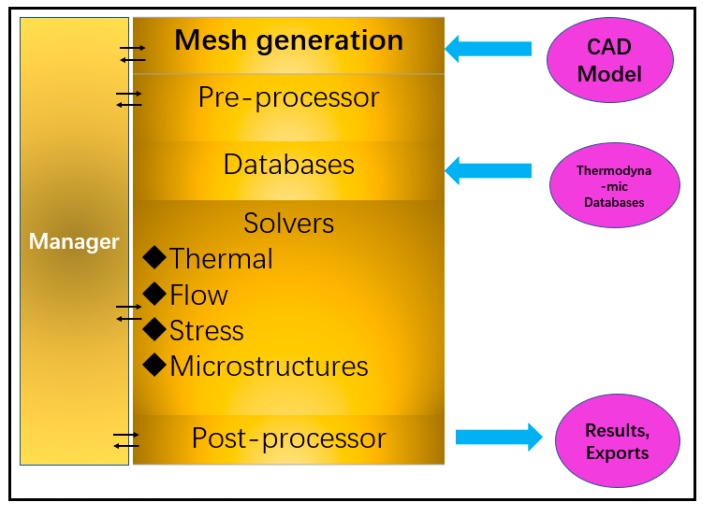
Structure of macro-model ProCAST.

**Figure 4 materials-12-03829-f004:**
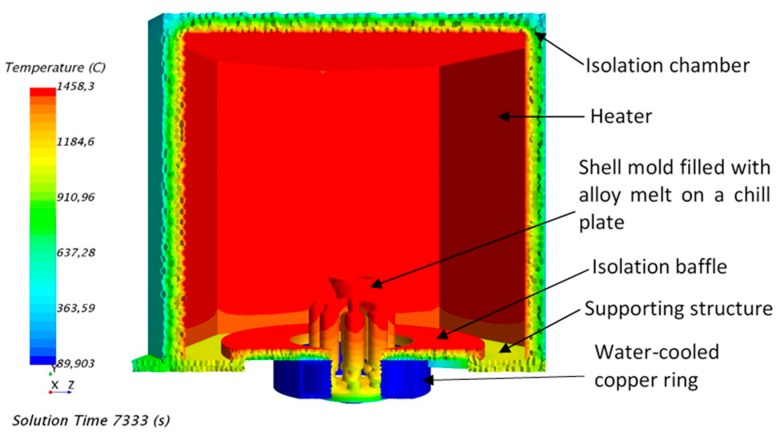
The simulation models of the Bridgman furnace.

**Figure 5 materials-12-03829-f005:**
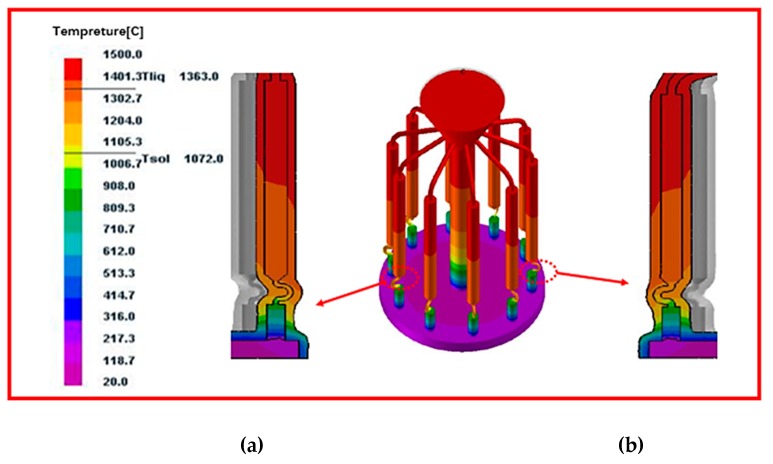
Simulation results of temperature field distribution and the morphology of solid-liquid interface of grain selector with different diameters without enclosure and the cross-section of (**a**) d_w_ = 2.4 mm; and (**b**) d_w_ = 4 mm.

**Figure 6 materials-12-03829-f006:**
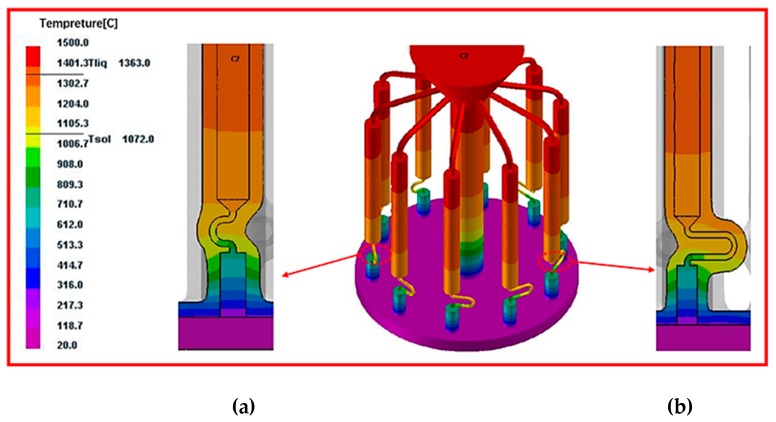
Simulation results of temperature field distribution and the morphology of solid-liquid interface of grain selector with different pitch length without enclosure and the cross-section of (**a**) d_s_ = 4 mm and (**b**) d_s_ = 20 mm.

**Figure 7 materials-12-03829-f007:**
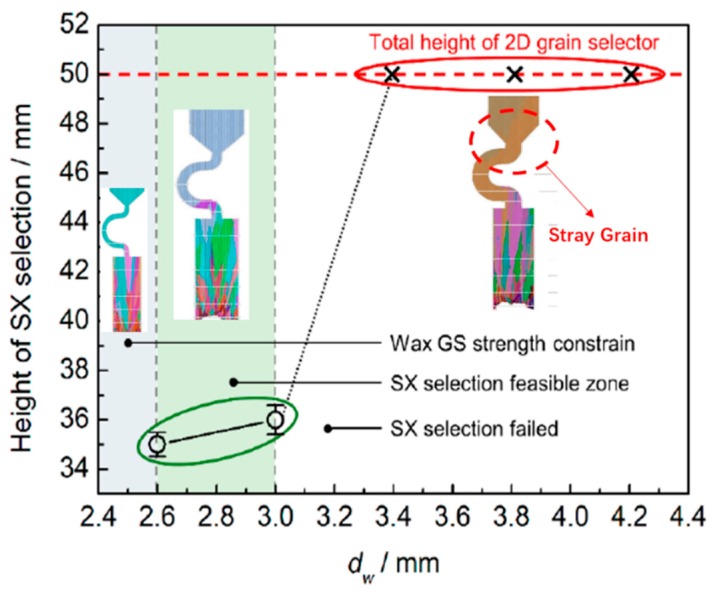
The simulation of different C-type selector diameter with related SX height.

**Figure 8 materials-12-03829-f008:**
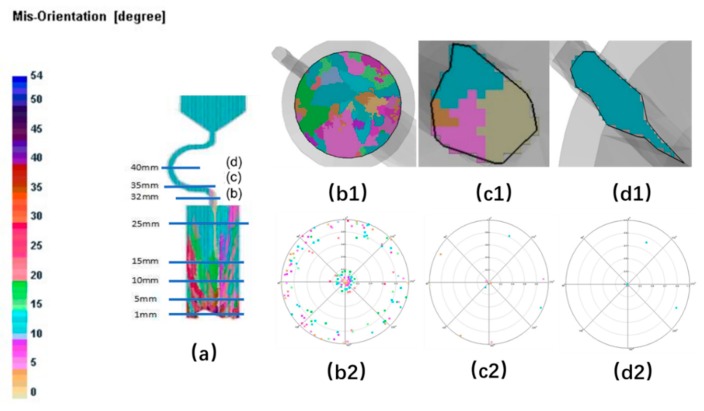
Simulation structure of C_1_ (2.4 mm) (**a**); EBSD (Electron Back Scattered Diffraction) maps of the grain structure evolution (**b1**,**c1**,**d1**) and the EBSD inverse pole figures (**b2**,**c2**,**d2**).

**Figure 9 materials-12-03829-f009:**
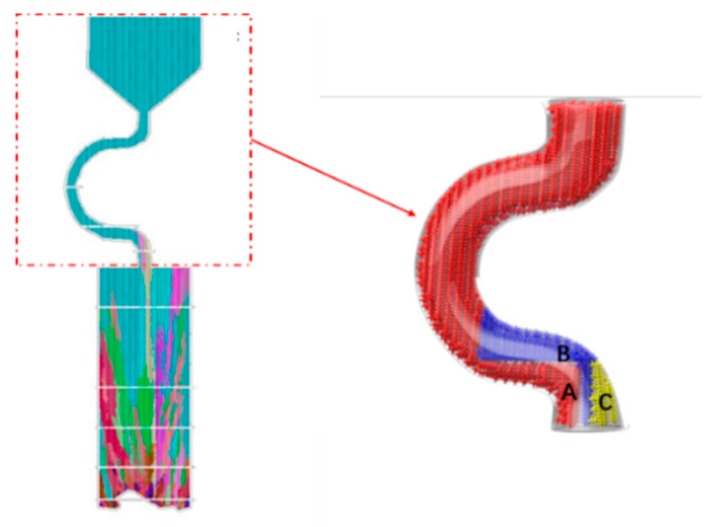
Simulation results of Grain A, B, C in the selector parts of C_1_: d_w_ = 2.4 mm, d_s_ = 8 mm.

**Figure 10 materials-12-03829-f010:**
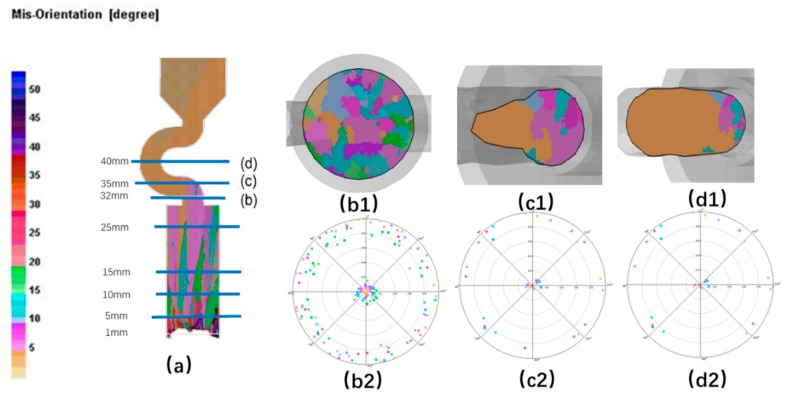
Simulation structure of C_4_ (4 mm) (**a**); EBSD maps of the grain structure evolution (**b1**,**c1**,**d1**) and the EBSD inverse pole figures (**b2**,**c2**,**d2**).

**Figure 11 materials-12-03829-f011:**
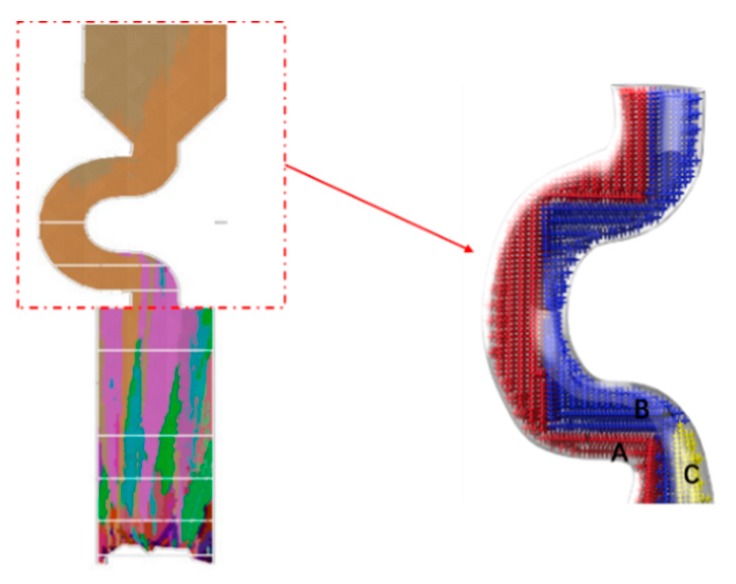
Simulation results in the selector parts of C_4_ (d_w_ = 4 mm, d_s_ = 8 mm).

**Figure 12 materials-12-03829-f012:**
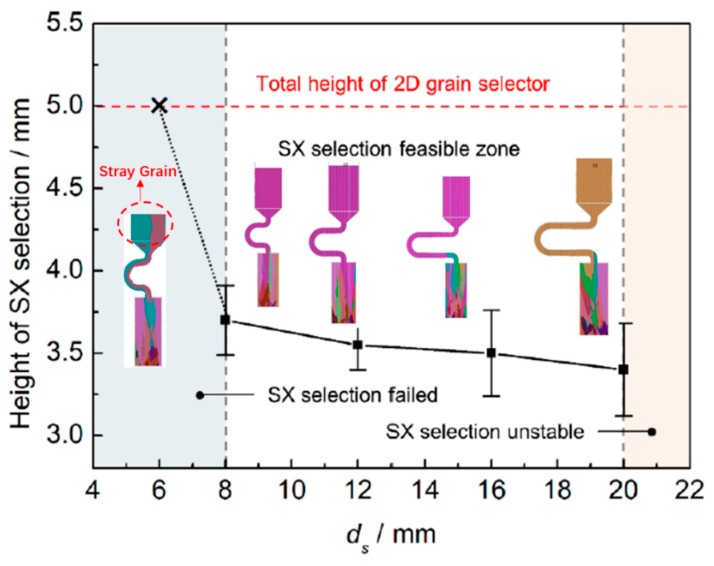
The simulation of different pitch length with related SX height.

**Figure 13 materials-12-03829-f013:**
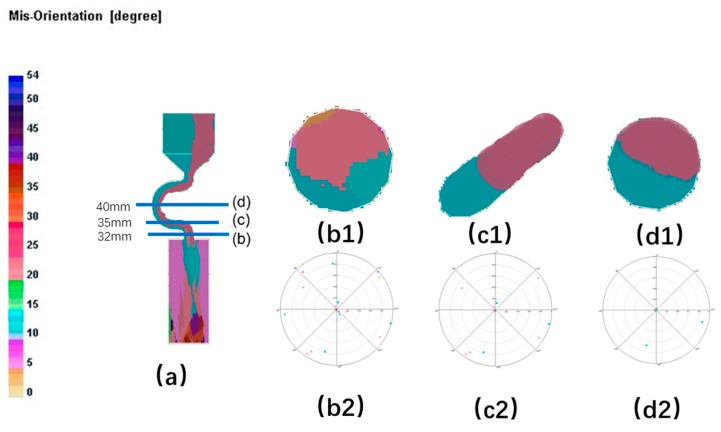
Simulation structure of C_1_ (4 mm) (**a**); EBSD maps of the grain structure evolution (**b1**,**c1**,**d1**) and the EBSD inverse pole figures (**b2**,**c2**,**d2**).

**Figure 14 materials-12-03829-f014:**
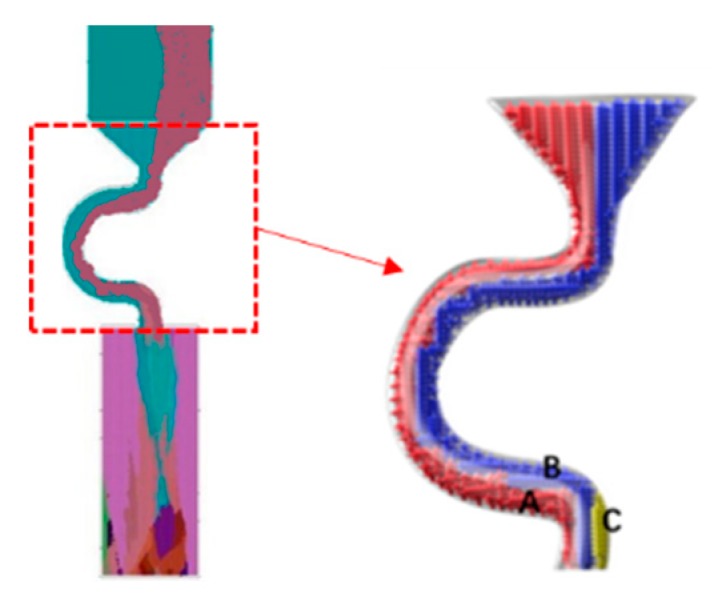
Simulation results in the selector parts of C_1_ (d_w_ = 3 mm, d_s_ = 4 mm).

**Figure 15 materials-12-03829-f015:**
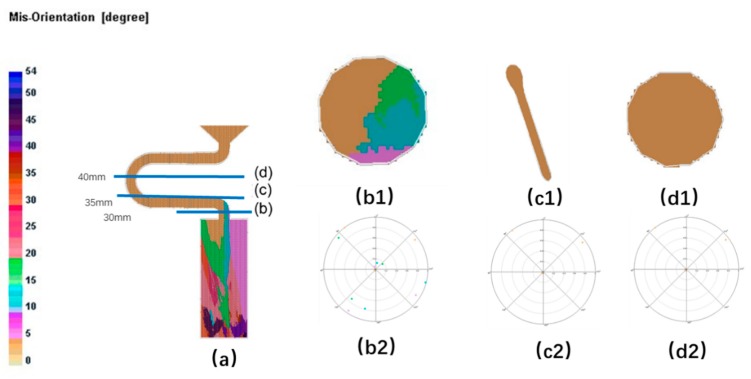
Simulation structure of C_5_ (20 mm) (**a**); EBSD maps of the grain structure evolution (**b1**,**c1**,**d1**) and the EBSD inverse pole figures (**b2**,**c2**,**d2**).

**Figure 16 materials-12-03829-f016:**
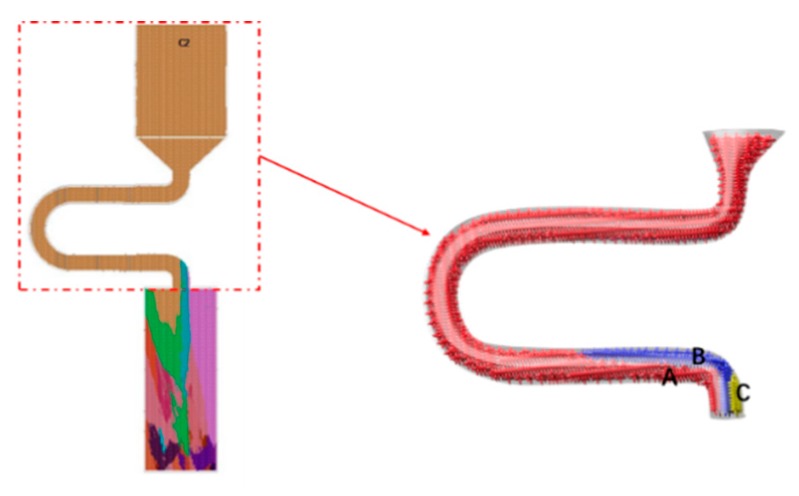
Simulation results in the selector parts of C_5_ (d_w_ = 3 mm, d_s_ =20 mm).

**Figure 17 materials-12-03829-f017:**
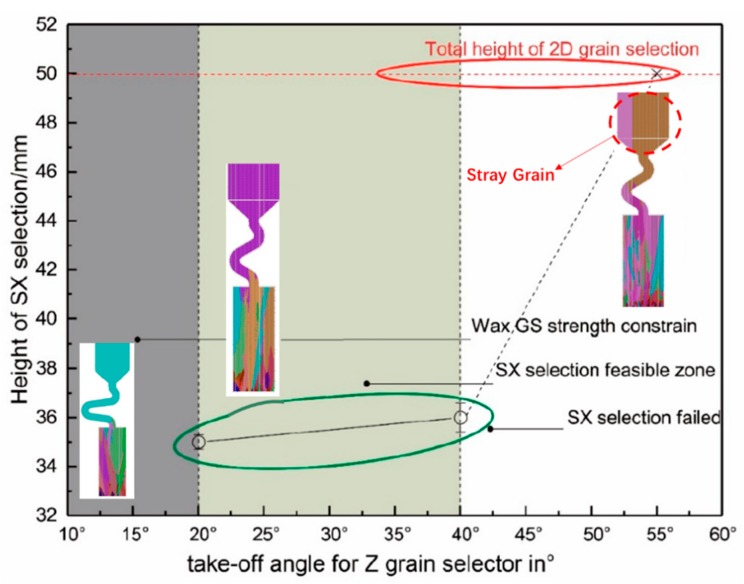
The simulation of different Z-type selector take-off angle with related SX height.

**Figure 18 materials-12-03829-f018:**
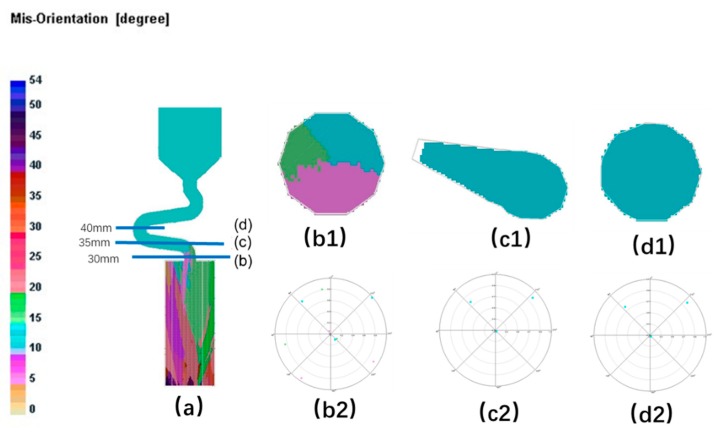
Simulation structure of Z_1_ (15°) (**a**); EBSD maps of the grain structure evolution (**b1**,**c1**,**d1**) and the EBSD inverse pole figures (**b2**,**c2**,**d2**).

**Figure 19 materials-12-03829-f019:**
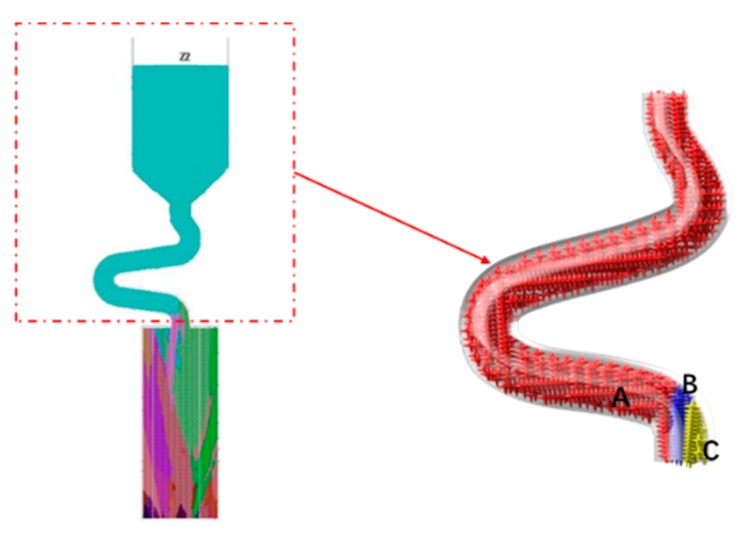
Simulation results in the selector parts of Z_1_ (15°) take-off angle.

**Figure 20 materials-12-03829-f020:**
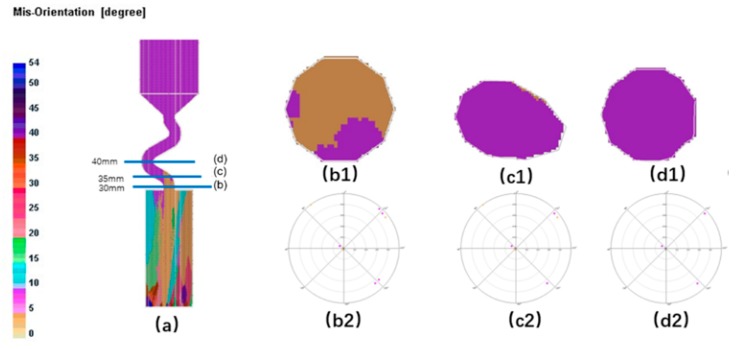
Simulation structure of Z_3_ (40°) (**a**); EBSD maps of the grain structure evolution (**b1**,**c1**,**d1**) and the EBSD inverse pole figures (**b2**,**c2**,**d2**).

**Figure 21 materials-12-03829-f021:**
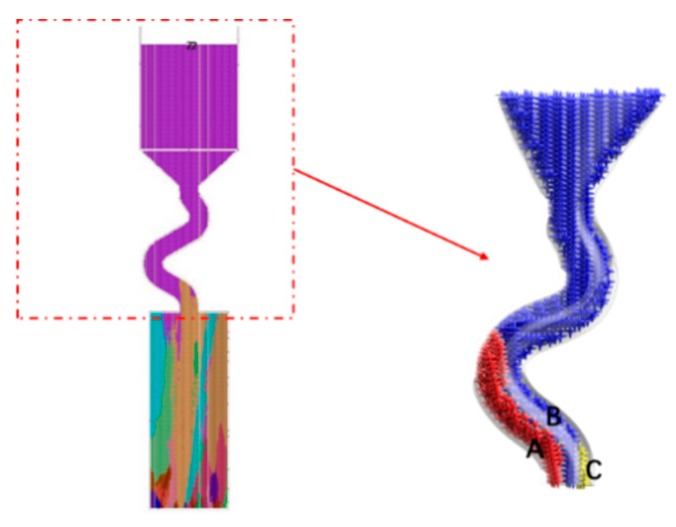
Simulation results of Z-shape grain selector with Z_3_ (40°) take-off angle.

**Table 1 materials-12-03829-t001:** The chemical composition of superalloy CM247LC (wt. %).

Elements	Al	Ti	Cr	Mo	Co	W	Ta	Hf	C	Ni
Wt. %	5.49	0.74	8.03	0.5	9.41	9.87	2.9	1.36	0.094	Bal.

**Table 2 materials-12-03829-t002:** Variation in thickness of the grain selector portion.

C-Form Grain Selector with a Variant Diameter	Single Crystal		Stray Grain	
	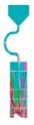	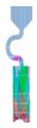	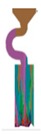	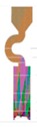
Stray Grain	No		Yes	
Probe	C_1_	C_2_	C_3_	C_4_
Diameter (mm)	2.4	3	3.4	4

**Table 3 materials-12-03829-t003:** Variant pitch length parameters of C-form selectors.

C-Form Grain Selector with a Variant Pitch Length	Stray Grain	Single Crystal			
	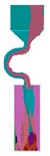	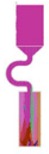	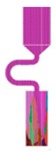	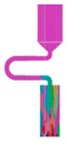	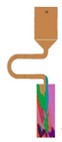
Stray Grain	Yes	No			
Probe	C_1_	C_2_	C_3_	C_4_	C_5_
Pitch length(mm)	4	8	12	16	20

**Table 4 materials-12-03829-t004:** Variant take-off angle parameters of Z-form selectors.

Z-Form Grain Selector with a Variant Take-Off Angle	Single Crystal			Stray Grain
	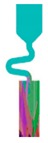	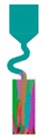	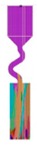	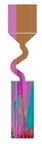
Stray Grain	No			Yes
Probe	Z_1_	Z_2_	Z_3_	Z_4_
Take-off angle	15°	30°	40°	55°

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
