# Peer review of "2-D Selector Simulation Studies on Grain Selection for Single Crystal Superalloy of CM247LC"

_materials, 2019, doi:10.3390/ma12233829_

Round 1
Reviewer 1 Report
General comments:
Other studies on grain selection, for example those cited below, have reported 3D simulations:
Effect of spiral shape on grain selection during
casting of single crystal turbine blades
J. Dai, J.-C. Gebelin, N. D'Souza, P. D. Brown & H. B. Dong
To cite this article: H. J. Dai, J.-C. Gebelin, N. D'Souza, P. D. Brown & H. B. Dong (2009) Effect of
spiral shape on grain selection during casting of single crystal turbine blades, International Journal
of Cast Metals Research, 22:1-4, 54-57, DOI: 10.1179/136404609X367317
To link to this article: https://doi.org/10.1179/136404609X367317
Grain Selection in Spiral Selectors During Investment Casting
of Single-Crystal Components: Part II. Numerical Modeling
H.J. DAI, H.B. DONG, N. D’SOUZA, J.-C. GEBELIN, and R.C. REED
METALLURGICAL AND MATERIALS TRANSACTIONS A VOLUME 42A, NOVEMBER 2011—3439
Simulation and Experimental Studies on Grain
Selection and Structure Design of the Spiral Selector
for Casting Single Crystal Ni-Based Superalloy
Hang Zhang 1 and Qingyan Xu 2,*
Materials 2017, 10, 1236; doi:10.3390/ma10111236 www.mdpi.com/journal/materials
The authors interchange the terms 2D and 3D. For example, in lines 34 and 37 the grain selectors are 3D. Simulations could be in 2D.
Please give more detail on how you reach the optimised condition given in the conclusion section as this has not been explained in the manuscript.
The grain selector’s take-off angle with 40°, a diameter of 3 mm, and pitch length of 8mm, respectively, are recommended.
Perhaps a Table containing the design parameters from the literature described in the Introduction and their effects on the single crystal growth.
Minor comments:
Please clearly distinguish though the manuscript between experimental and simulations and what will be reported in the corresponding sections. For example:
Title: 2-D Selector Simulation Studies…
Abstract: “By using directional experiments and the cellular Automaton Finite Element (CAFE)…
Section 3 title: Simulation results and discussion is not appropriate as EBSD results are also reported (Figs. 8, 10, 13, 15, 18, 20
Lines 14-15:
Please specify the dimensions of the grain selector geometries investigated (optional suggestion).
Line 20: Please define SC
References needed, for examples:
Lines 27-28:
The geometry of the selector determines the grain selecting efficiency [Reference(s)?].
Lines 38-40:
Previous studies have shown that the geometric parameters of a spiral grain selector (i.e., the parameters of the starter block and selection block) directly determine the final result of the selected crystal [Reference(s)?].
Lines 45-47: “The height of the grain selector and the diameter of the starter block have no significant effect on the final acceptability of the single crystal.”
Some references are not properly cited. For example, in lines 63 and 80, Dong et al. and Dai, respectively.
Lines 28-30: “The previous investigation indicates that the smaller wire diameter, the spiral selector becomes more efficient. However, if the selector diameter is too small, the spiral selector is not strong enough to support the SC casting.”
The two above sentences are contradictory. Please rewrite or give more details.
Line 35: “The variation in the geometry of the different materials was studied”.
Not enough detail is given.
Please enlarge grain selector images, for example in Tables 2 and 3 they could be enlarged closer to the box frames.
Please input geometry parameters in Tables 2, 3 and 4
Please specify the picture in line 158 and figure in line 165?
What image correspond to 4mm abd 2.4mm in Fig. 5.
Temperature numbers in Figs. 5 and 6 are difficult to read
Lines 211-212. The summary is very brief.
Line 220: “undercoolingError! Reference source not found” ??
Author Response
Response to Reviewer 1 Comments
Point 1: Line 20: Please define SC
Response 1: Single Crystal components are conventionally produced by using the directional solidification process.
Point 2: Lines 27-28: The geometry of the selector determines the grain selecting efficiency [Reference(s)?].
Response 2: The geometry of the selector determines the grain selecting efficiency [6].
Point 3: Lines 38-40: Previous studies have shown that the geometric parameters of a spiral grain selector (i.e., the parameters of the starter block and selection block) directly determine the final result of the selected crystal [Reference(s)?].
Response 3: Previous studies have shown that the geometric parameters of a spiral grain selector (i.e., the parameters of the starter block and selection block) directly determine the final result of the selected crystal [9].
Point 4: Lines 45-47: “The height of the grain selector and the diameter of the starter block have no significant effect on the final acceptability of the single crystal.” [Reference(s)?]
Response 4: The height of the grain selector and the diameter of the starter block have no significant effect on the final acceptability of the single crystal [10].
Point 5: Lines 28-30: “The previous investigation indicates that the smaller wire diameter, the spiral selector becomes more efficient. However, if the selector diameter is too small, the spiral selector is not strong enough to support the SC casting.”
The two above sentences are contradictory. Please rewrite or give more details.
Response 5: The previous investigation indicates that the smaller wire diameter, the spiral selector becomes more efficient. However, if the selector diameter is too small, the spiral is too thin and fragile, not strong enough to support the SC casting.
Point 6: Line 35: “The variation in the geometry of the different materials was studied”.
Not enough detail is given.
Response 6: The variation in the geometry of the different materials was studied [7,8].
Point 7: Please enlarge grain selector images, for example in Tables 2 and 3 they could be enlarged closer to the box frames.
Response 7: The selector image size has been adjusted respectively.
Point 8: Please input geometry parameters in Tables 2, 3 and 4
Response 8: The related geometry parameters have been input.
Point 9: Please specify the picture in line 158 and figure in line 165?
Response 9: As shown in the picture 4, the cooling efficiency decreases with the distance from the cooling plate. The following figures(Figure.5 and Figure.6) shows the temperature field distribution and solid-liquid interface morphology at different withdrawal times under the withdrawal rate of 3mm/min.
Point 10: What image correspond to 4mm abd 2.4mm in Fig. 5.
Response 10: The figure size has been adjusted respectively.
Point 11: Temperature numbers in Figs. 5 and 6 are difficult to read
Response 11: The figure size has been adjusted respectively.
Point 12: Lines 211-212. The summary is very brief.
Response 12: Although the results show that the effect of selecting the smaller diameter selector is better, however, since when the diameter is smaller than 2.4mm, the selector is in low strength and is easy to deform, and the diameter of 3mm is expected considering the stability. To sum up, 3mm is the optimal diameter.
Point 13: Line 220: “undercoolingError! Reference source not found” ??
Response 13: Like the case of selector diameter, at the beginning of the solidification, grain B grows faster than grain A and grain C due to the good alignment with the <001> direction and the lower required undercooling.
Reviewer 2 Report
Review
The submitted manuscript is a substantial extension of authors’ scientific papers published in Materials 2019, 12, 789 (references no 3 of current manuscript) and Materials 2019, 12, 1781 (not cited in current manuscript). This proves that the topic itself fulfils the requirements of the journal. In general the manuscript is based on the previously described idea, but, in opinion of reviewer, the described simulations of the selected superalloy provide enough new data to judge publishing of the subsequent manuscript of the series. Because manuscript possesses some minor errors (described below), the reviewer cannot recommend publication of the manuscript in the current form and suggests revision.
Comments
The reference to closely related authors’ scientific paper (published recently in Materials 2019, 12, 1781) should be added.
The following closely related paper and basic review concerning studied topic: Hassan Gheisari, Ebrahim Karamian, Journal of Modern Processes in Manufacturing and Production, Vol.4, No.1, Winter 2015; M. Rappaz, Ch.-A. Gandin, A. M. Stoneham, M. McLean and M. S. Loveday, Philosophical Transactions: Physical Sciences and Engineering, Vol. 351, No. 1697, High-Temperature Structural Materials (Jun. 15, 1995), pp. 563-577, should be also referred in the manuscript.
All used abbreviations should be explained upon first use (e.g SC on the first page).
Use the C1 (in variant pitch length) and Z4 probe symbol for stray grains versus C4 (in variant pitch length) and Z1 for singe crystals is misleading and cause manuscript harder to follow. Reviewer recommend inverse of numbering order for probes Z.
The Figures 5 and 6 should have the same size to allow easy comparison. Additionally they should be combined into one figure.
In all cases the pole figures are too small to be readable.
The Figures 8 and 10 should be combined into one figure.
The Figures 9 and 11 should be combined into one figure.
The Figures 13 and 15 should be combined into one figure.
The Figures 14 and 16 should be combined into one figure.
The Figures 18 and 20 should be combined into one figure.
The Figures 19 and 21 should be combined into one figure.
The language of manuscript should be carefully checked as in current form it contains soma awkward phrases, e.g. “The main steps are divided into several steps” (which should be e.g. “The main steps are divided into several sub-steps”); “The starter is composed of parameters of 10 mm (L)×10 mm (W)×30 mm (H)” (which should be e.g. “The starter size is 10 mm (L)×10 mm (W)×30 mm (H)”).
Author Response
Response to Reviewer 2 Comments
Point 1: All used abbreviations should be explained upon first use (e.g SC on the first page).
Response 1: Single Crystal components are conventionally produced by using the directional solidification process.
Point 2: The Figures 5 and 6 should have the same size to allow easy comparison. Additionally they should be combined into one figure.
Response 2: The figure size has been adjusted respectively.
Point 3: In all cases the pole figures are too small to be readable.
Response 3: The pole figure size has been adjusted respectively.
Point 4: The Figures 8 and 10 should be combined into one figure.
The Figures 9 and 11 should be combined into one figure.
The Figures 13 and 15 should be combined into one figure.
The Figures 14 and 16 should be combined into one figure.
The Figures 18 and 20 should be combined into one figure.
The Figures 19 and 21 should be combined into one figure.
Response 4: The figures have been combined and adjusted respectively.
Point 5: The language of manuscript should be carefully checked as in current form it contains soma awkward phrases, e.g. “The main steps are divided into several steps” (which should be e.g. “The main steps are divided into several sub-steps”); “The starter is composed of parameters of 10 mm (L)×10 mm (W)×30 mm (H)” (which should be e.g. “The starter size is 10 mm (L)×10 mm (W)×30 mm (H)”).
Response 5: The main steps are divided into several sub-steps.
The starter size is 10 mm (L)×10 mm (W)×30 mm (H).
Reviewer 3 Report
While the paper has a very specific audience it is useful from the point of view of what is available, it's approach and the soundness/completeness of the investigation. It was well researched and written.
Author Response
Response to Reviewer 3 Comments
Point 1: While the paper has a very specific audience it is useful from the point of view of what is available, it's approach and the soundness/completeness of the investigation. It was well researched and written.
Response 1: Very appreciate for you kind review and comments.